# From Silver Nanoflakes to Silver Nanonets: An Effective Trade-Off between Conductivity and Stretchability of Flexible Electrodes

**DOI:** 10.3390/ma12244218

**Published:** 2019-12-16

**Authors:** Liqiao Chen, Zhe Leng, Yunqian Long, Xuan Yu, Wei Jun, Xiaoming Yu

**Affiliations:** 1Institute of Innovation & Application, Zhejiang Ocean University, Zhoushan 316022, China; chenlq118@zjou.edu.cn (L.C.); lengzhe12345a@163.com (Z.L.); longyunqian@163.com (Y.L.); yuxuan@zjou.edu.cn (X.Y.); 2National Engineering Research Center for Marine Aquaculture, Zhejiang Ocean University, Zhoushan 316022, China; 3School of Naval Architecture and Mechatronics Engineering, Zhejiang Ocean University, Zhoushan 316022, China; johnweijun@163.com

**Keywords:** Ag nanoflakes, Ag nanonets, solvothermal transformation, bending conductivity, elastic electrodes

## Abstract

Flexible and stretchable conductive materials have received significant attention due to their numerous potential applications in flexible printed electronics. In this paper, we describe a new type of conductive filler for flexible electrodes—silver nanonets prepared through the “dissolution–recrystallization” solvothermal route from porous silver nanoflakes. These new silver fillers show characteristics of both nanoflakes and nanoparticles with propensity to form interpenetrating polymer–silver networks. This effectively minimizes trade-off between composite electrode conductivity and stretchability and enables fabrication of the flexible electrodes simultaneously exhibiting high conductivity and mechanical durability. For example, an electrode with uniform, networked silver structure from the flakiest silver particles showed the lowest increase of resistivity upon extension (3500%), compared to that of the electrode filled with less flaky (3D) particles (>50,000%).

## 1. Introduction

Flexible printed electronics aim to minimize material waste and production costs, which was considered as an effective way to reduce carbon dioxide emissions during manufacturing [1,2]. With the development of nanomaterials and nanotechnology, various kinds of printing pastes ranging from conductors, insulators, and semiconductors comprised of nanostructures have been developed and utilized for solar cells [3], touch screens [4], transistors [5], sensors [6,7], and elastic/flexible devices [8,9]. Additionally, flexibility and stretchable properties are considered as the key parameters of flexible devices [10]. Moreover, the elastic electrodes as an important component are also necessary for future applications of flexible printed electronics.

Different micro/nano structures silver powders including Ag flakes and Ag nanoparticles were introduced to fabricate flexible electrodes [10]. The structures of the flake-shaped silver powders are some complete integral unit, which cannot form some “interpenetrating” nanostructure with the resin in the electrode layer, which limits its toughening function [11]. For the case of nanoparticles, the connecting way of “point-to-point” among the nanoparticles in the electrode layer arises high resistance, and rapid increasing of resistance then stretching happens. In spite of great effort, the trade-off of mechanical and electrical characteristics still remains in currently existing stretchable conductive materials [2].

The silver nanowires conductive network is a good choice to address the dilemma in conductivity and stretchability trade-offs [12]. Very recently, a paper dealt with the importance of stress release and behavior of Ag nanoparticles in flexible solar cell devices [13]. However, to obtain good conductivity and stretchability, further treatment for the silver nanowires network to form electrical and mechanical junctions among different nanowires is necessary. Welding the nanowire junctions is an effective strategy for reducing the sheet resistance and improving the operational stability of flexible nanowire electrode in practical applications [12,14]. Chemically growing silver “solder” or the chemical treatment for the junctions of the nanowires can also contribute to low-cost and highly stable electrodes [15,16]. This processing not only increases the preparation complexity of conducting film, but also imposes more limitations on the performance of the resin materials which are curing agents to keep its mechanical strength in the flexible electrode [10,13]. Therefore, the conductive fillers (silver powders) for the flexible silver electrode preferably simultaneously have double characteristics with the high conductivity of flakes and stretchability of nanoparticles [17,18].

Here, we directly synthesized silver nanonets powders as conductive fillers of silver electrodes instead of the silver nanowires conductive network. Firstly, we adopted a simple route to synthesize Ag porous nanoflakes composed of nanoparticles without using any capping agents. Interestingly, Ag nanoflakes could also convert into different morphologies such as irregular nanoparticles or porous nanonets only through the “dissolution–recrystallization” solvothermal process. We demonstrate the feasibility of nanoflakes as conducting fillers to fabricate elastic electrodes. Surprisingly, porous Ag nets simultaneously exhibit high conductivity and mechanical durability. This is probably due to the forming of the intercross and interpenetration structure between resins and Ag nanonets.

## 2. Experimental

### 2.1. Materials and Chemicals

Synthesis of typical silver nanoflakes. 8.0 mmol of AgNO_3_ was dissolved in 2.5 mL of water. 19.0 mmol of L-ascorbic acid was dissolved in 2.5 mL ethanol and 5.0 mL water under vigorous magnetic stirring. Additionally, 0.1 mmol of D-saccharic acid 1,4-lactone monohydrate was added into the ascorbic acid solution to prepare a reductive solution. Then, AgNO_3_ solution was quickly added into the reductive solution with vigorous stirring. After 5 min, a solid product was obtained, which was washed with distilled water and alcohol several times.

Solvothermal transformation of typical silver flakes. First, the above samples were transferred to a Teflon-lined stainless-steel autoclave (25 mL) and sealed after adding 15.0 mL H_2_O or another solvent. Solvothermal processing was carried out at 150 °C for different time. The resulting products were washed with distilled water and alcohol several times. Finally, the product was dried at 60 °C for 10 h.

### 2.2. Characterization and Methods

The products were characterized by powder XRD, electron microscopy SEM, and TEM. XRD patterns were recorded using a Rigaku D/MAX 2200 VPC diffractometer (Rigku, Tokyo, Japan) using CuK*α* radiation (λ = 0.15045 nm) and a graphite monochromator. SEM images were recorded using a Phenom-World scanning electron microscope (Phenom-World, Eindhoven, The Netherlands). TEM images were recorded using a JEM-2010HR transmission electron microscope (JEOL Ltd., Tokyo, Japan) operated at an accelerating voltage of 200 kV. TEM samples were prepared by dispersing the powders on a holey carbon film supported on copper grids.

The preparation and the evaluation of elastic conductors. Firstly, the Dynapol L210 rubber (Evonik, Marl, Germany) (10 wt%) was dissolved in the DBE as solvent by a magnetic stirrer at 60 °C for 2 h, then surfactant BYK-163 (BYK Additives (Shanghai) Co., Ltd., Shanghai, China) was added to obtain an adhesive agent, which was mixed with the as-prepared silver powders (70 wt%), and processed by a three-roller mill into the elastic conductive paste. Next, the paste was printed on the neoprene rubber sheet with a 15 mm width 160 mm length, and 1 mm thick using 125 μm-thick stainless steel shadow masks. After printing a 5 × 100 mm^2^ rectangular silver conductive film, the samples were dried at 150 °C for 30 min in air to remove excess solvent. The conductivity was measured using the four-probes-method for different bending and the elongation cycle times. Four typical samples were evaluated for each condition of the elastic conductor paste.

## 3. Results and Discussion

The powder X-ray diffraction patterns (XRD) of the typical synthesized sample are shown in Figure 1a. This matches well with JCPDS card No. 04-0783 for cubic Ag with a = b = c = 0.4086 nm, confirming that all the products were cubic-structure Ag. The SEM images of the samples (Figure 1b) show that Ag nanoflakes are with a width and thickness of approximately 10 μm and 60 nm, respectively, and dozens of nanometer-scale holes on its surface. The TEM images show that those holes are irregular and the sizes are about tens of nanometers (Figure 1c). The structures of these nanoflakes are also confirmed through HRTEM (Figure 1d). The lattice-resolved image taken on a single Ag nanoflake reveals fringes with separated spacing of 0.25 nm, being assigned to the Ag (1/3) {422} reflection, which has also been previously observed for Ag nanoplates, the assignment is consistent with the geometrical model, in which each triangular nanoplate is bound by two {111} planes as the top and bottom faces, and three {100} planes as the side faces [19].

The as-obtained silver powder here are with a rough surface and a large surface area, and can hardly be infiltrated with the organic adhesive. In the process of preparing the elastic conductor paste, silver powder is rolled into a millimeter-scale piece, and the original structure of the silver powder is destroyed. Therefore, it is necessary to firstly reduce the high activity point of silver powder to make the silver powder fully infiltrated with the organic binder to form a uniform conductive paste.

Some nanostructures are unstable and they possibly change under solvothermal conditions as previously reported [20]. Thus, we investigated the evolution of the structure of Ag nanoflakes under different solvothermal processing. The as-obtained Ag nanoflakes (2.0 g) were differently added to 10 mL of distilled water, 0.01 mM HCL/water, 0.01 mM NaOH/water, and diethylene glycol aging at 150 °C for 6 h. As shown in the SEM images in Figure 2a–d, the silver powders become an irregular shape, and the sizes of nanoflakes decrease, and their thickness increased after aging in pure water for 6 h. In the condition of pH = 4, the as-synthesized products show an irregular and spherical shape. When increased, the pH of the reaction solution to alkaline (pH = 10), and some irregular shape with smaller particle sizes could be obtained. Ag flakes reduce to 2 μm and their thickness increased to 126 nm, and the sizes of the cores in nanoflakes increased when diethylene glycol was used instead of water (Figure 2d). Regardless of the different conditions, silver nanoflakes have the same tendency to evolve from two-dimensional to three-dimensional structures. Therefore, it is very meaningful to further process the silver nanoflakes into the different structures by adjusting the aging time.

In our previous paper, we reported that α-Fe_2_O_3_ flakes could gradually transform into nanoparticles through the “dissolution–recrystallization” solvothermal process and “a mass transport” took place [20]. Herein, the experiments showed that Ag nanoflakes have similar features. Further study on the structural evolution mechanism of Ag nanoflakes might help us have a better understanding about the physical process by monitoring the evolution of Ag flakes at different times.

The as-synthesized Ag nanoflakes (2.0 g) were added to 10 mL of distilled water. The growth process of this route was continuously monitored by collecting the samples from different aging times. The morphology was characterized by SEM (Figure 3). As shown in Figure 3a, the density of holes in the Ag flakes decrease, and the diameters of holes increase from 100 to 200 nm when the particles aged for 0.5 h, also, the thickness of flakes increase. If prolonging the aging time to 4 h, the size of the flakes and the number of holes decrease (Figure 3b). By increasing the aging time to 8 h, the flake-like structures almost disappear, and irregular shapes such as biscuits are obtained (Figure 3c). All the holes vanish, and the size of the products reduced to less than 2 µm when the aging time is extended to 24 h.

The evolution process of Ag nanostructures could be illustrated in Figure 4. First, the edges of Ag nanonets and their holes gradually dissolve. The holes in silver nets grow bigger and bigger, and several small holes merge into larger holes. This trend leads to the initial one where the large piece finally breaks into several small fragments. According to “Ostwald ripping” mechanism [21], that is, the small particles gradually dissolve due to the high surface energy, and the large particles continue to grow. Continuous mass transport and growth from two dimensions to the three dimensions ultimately evolve into a stable structure with a low surface energy, usually 3D polyhedrons [20].

Further, we prepared four conductive pastes using four kinds of silver powders in Figure 3 as conductive fillers and compared their flexibility of the four samples, respectively. The four conductors were obtained when the four conductive pastes that were printed were heated for 30 min and named for sample #1, sample #2, sample #3, and sample #4. According to the results shown in Figure 5a, the resistance of four samples all increased when the conductors were stretched, but the increasing value of sample #1 was obviously smaller than that of the other three samples. For example, the elongation of conductor for sample #1 was 140%, and the resistance increased by 3500%, but for sample #4 the lost conductive function was at the same elongation. Therefore, the silver powders with more uniform, net-structure significantly improves the stretching properties of the conductor. The resistance changes of the four samples after bending for multiple cycles are also measured. The results are shown in Figure 5b. Similarly, sample #1 showed more excellent bending conductivity than the other samples. We believe that the four samples prepared in this paper do not bring out their best performance. Further surface modification of the silver nanonets and optimization of agents’ selection, such as the selection of rubbers and the surfactant [1] can promote the formation of a better network interpenetrating structure between the silver powders and the resins. In this way, the advantages of the special structure silver nanonets could be fully exerted. Much efforts and optimized experiments would be necessary and ongoing.

The reason that enhanced bending and elongation conductivity are simultaneously realized by the introduction of the nanonets as conductive fillers can be attributed to their special structures. The deformation of conductive networks during stretching or bending could destroy the conductive channel in conductors. The schematic diagram was shown in Figure 6. When Ag nets were used as the conductive fillers, Ag nets are uniformly dispersed in a rubber matrix, and the Ag nets could form an intercross and interpenetration structure with resin where the resin or rubber molecular are connected with nanonets through holes to maintain mechanical durability, at the same time, conductive paths in silver conductive networks would have few influences after bending or stretching. The combination of conductivity and stretchability are enabled by a special conductive network structure of intercross and interpenetration between silver and polymer matrix in the printed elastic conductor [21]. For the case of nanoparticles or nanoflakes, the contact point or contact area reduced when stretching happened, which resulted in rapid increase of resistance.

In conclusion, we adopted a simple route to investigate the transformation process from flakes to nanonets using the “dissolution-recrystallization” process. The reaction system in this method is simple, and the samples obtained were dispersed with a clean surface because only water was added without adding any other surfactants or reagents. This route not only provided the convenience for getting novel Ag nanostructures, but also offered a special conductive filler to realize elastic conductors, which have been proven to be feasible as new electronic functional fillers that simultaneously exhibit high conductivity and mechanical durability in elastic electrodes.

## Figures and Tables

**Figure 1 materials-12-04218-f001:**
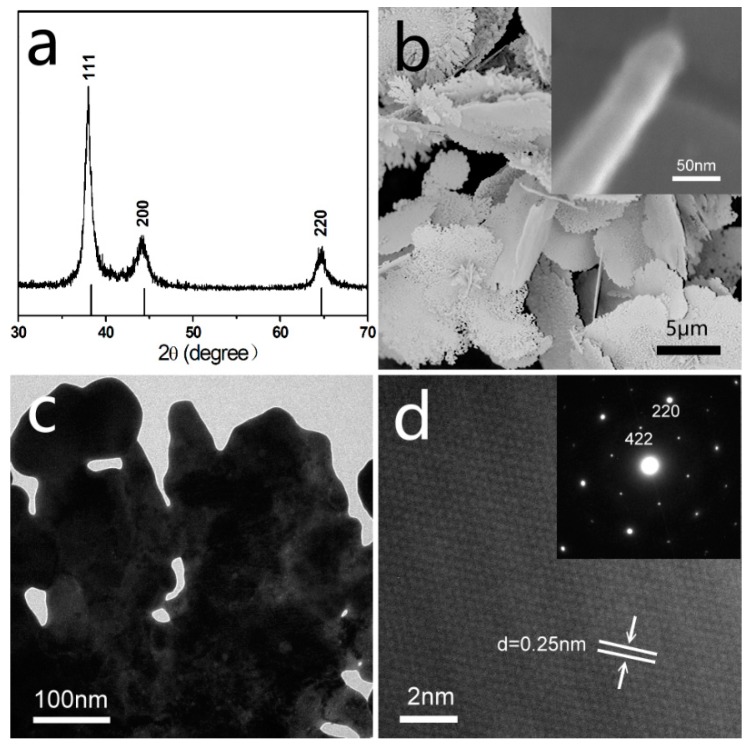
The structure and shape of the as-synthesized Ag nanoflakes, (**a**) powder XRD; (**b**) SEM images and the magnification (inset); (**c**) TEM images; (**d**) HRTEM (High Resolution Transmission Electron Microscope) images and SEAD (Selected Area Electron Diffraction) (inset).

**Figure 2 materials-12-04218-f002:**
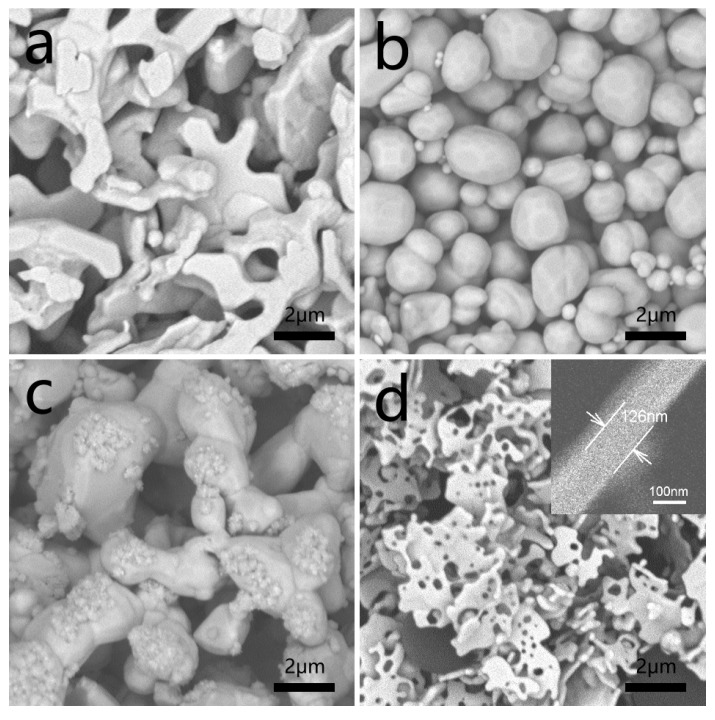
SEM images of the samples from the solvothermal evolution of Ag nanoflakes after aging at 150 °C for 6 h under different solvents, differently. (**a**) In DI water; (**b**) 0.01 mM HCl/DI water solution; (**c**) 0.01 mM NaOH/DI water solution; (**d**) diethylene glycol.

**Figure 3 materials-12-04218-f003:**
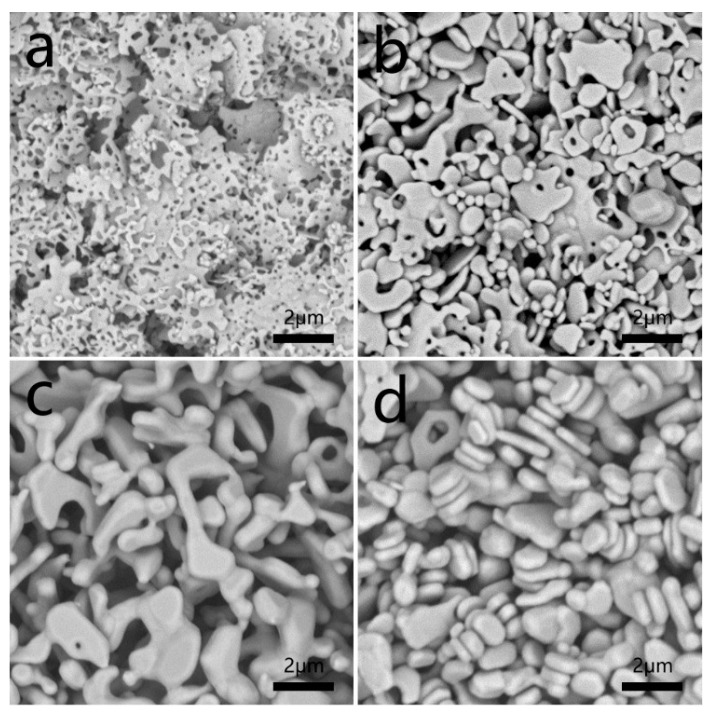
SEM images of the samples from the solvothermal evolution of Ag nanoflakes after aging at 150 °C in DI water for different aging time, differently. (**a**) 0.5 h; (**b**) 4 h; (**c**) 8 h, and (**d**) 24 h.

**Figure 4 materials-12-04218-f004:**
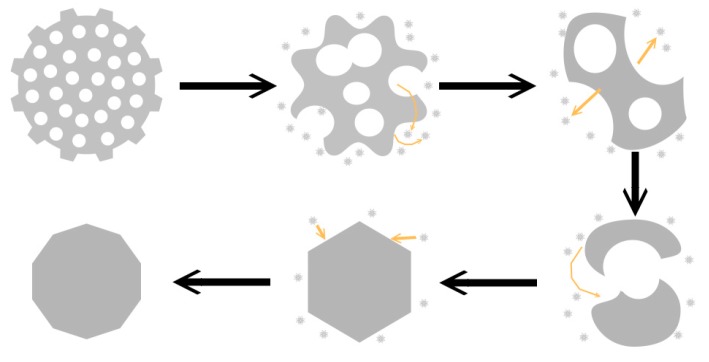
Schematic diagram of the structure evolution from Ag nanoflakes to Ag nanonets during the solvothermal evolution.

**Figure 5 materials-12-04218-f005:**
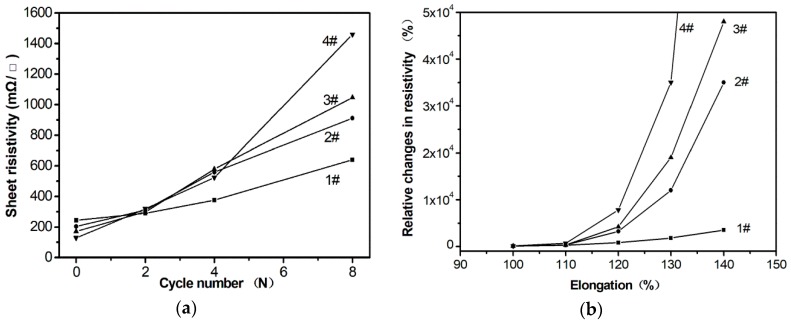
Sheet resistance of samples #1, #2, #3, #4 changes with bending cycle times (**a**) and the elongation (**b**).

**Figure 6 materials-12-04218-f006:**
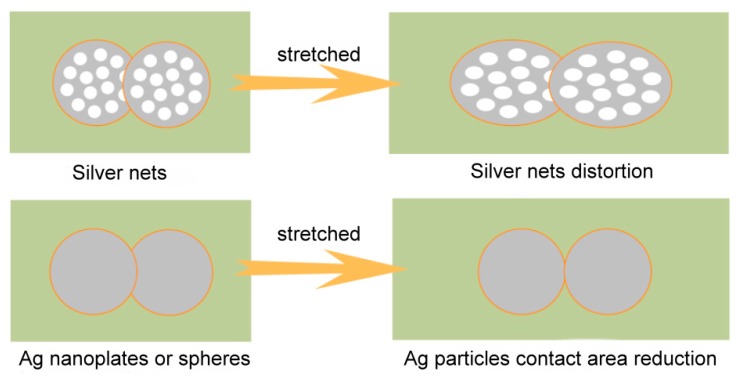
The schematic diagram for the change of contact areas between different structures of Ag conductive particles in elastic electrodes when stretching happened.

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
