# Peer review of "From Silver Nanoflakes to Silver Nanonets: An Effective Trade-Off between Conductivity and Stretchability of Flexible Electrodes"

_materials, 2019, doi:10.3390/ma12244218_

Round 1

Reviewer 1 Report

Authors presented their recipe of changing the morphology of silver from nanoflakes to nanomeshes to increase flexibility while maintaining acceptable conductivity.

Few corrections must be made and the revised version must be re-reviewed.

The paper is not well-written. It must be proofread and extensively modified by a person fluent in English and familiar with the technical writing in material science. In many places the proper non-technical words have been used. Authors present the SEM pictures of their sample aged in different solutions in figure 2. (a) By The look of the figures, it looks like figure 2-d has the closest morphology to nanomeshes and must lead to better stretchability. So why authors used DI water to proceed with the rest of the paper and why they did not use diethylene glycol instead? And (b) why they did not show the effect of different chemicals they used on resistivity and sheet resistance? In figure 4, the authors need to compare their results with a sample not aged at all to show that their method is advantageous in the first place. Specially because it looks like the best result belongs to the sample that is aged for a shortest time.

Author Response

The paper is not well-written. It must be proofread and extensively modified by a person fluent in English and familiar with the technical writing in material science. In many places the proper non-technical words have been used. Authors present the SEM pictures of their sample aged in different solutions in figure 2. (a) By The look of the figures, it looks like figure 2-d has the closest morphology to nanomeshes and must lead to better stretchability. So why authors used DI water to proceed with the rest of the paper and why they did not use diethylene glycol instead? And (b) why they did not show the effect of different chemicals they used on resistivity and sheet resistance? In figure 4, the authors need to compare their results with a sample not aged at all to show that their method is advantageous in the first place. Specially because it looks like the best result belongs to the sample that is aged for a shortest time. 

Response: The manuscript has been modified and polished again;

We used DI water to proceed with the rest of the paper and did not use diethylene glycol instead. In fact, we found that both water and alcohol could realize the same function, but DI water could short aging time, and is also more environmentally friendly to prepare product on a large scale later. The silver powder obtained by water treatment is the most representative structure, which could explain the relationship between the structure of silver powder and the electrical conductivity and flexibility of the film.

For the results with a sample not aged at all, we find silver powders can hardly be infiltrated with the organic system and rolled into a millimeter-scale piece in the process of preparing the elastic conductor paste which probably due to its rough surface and high surface area. Therefore, it is necessary to firstly reduce the high activity point of silver powder to make silver powder fully infiltrate with the organic binder to form a uniform conductive paste. We also add this part in the manuscript.

Reviewer 2 Report

First of all, This work was published before, please see

doi:10.20944/preprints201905.0375.v1

This study reported the silver nanonets processed that have the features of nanoflakes and nanoparticles from porous silver nanoflakes as conducting fillers of flexible conductive films.

There are some comments to be made and covered as follows:

1-      With respect to the abstract part, there is a lack of information and the novelty of the work needs more clarification. In addition, this part should be reviewed on the basis of the material, the objective, the main results obtained, as well as clarification of the work novelty.

2-      Keywords; the provided Keywords should be revised for example the author provids "ag flakes", moreover further keywords should be added related to the work tabout targeted, method or techniques, Ag nanonets......

3- The Moreover, the provided claim at the end of the introduction needs more clarification and revision on the basis of the objective of the study, the material and approach, the relevance, the expected, and the key findings.

4-      The experimental part:

i) can the author divide this part to "Materials and chemical” section

    , and “Characterization and methods”., and 

ii) The preparation method needs more clarification, in fact it is better to add scheme that clarify each step of preparation till the detection process. This means scheme 1 should be revised.

5-   In fact during the results and discussion, the author described the figures only without clear discussion or interpretation. Therefore, it is highly recommended to give more information and details about the results obtained.

6-     Moreover, the provided figure captions is not enough to describe the figure, the author must provide more information and details enough to clarify the figures.

7-      The author should provide all figures with high resolution.

  8-        Clear results should be provided in the conclusion part.

Author Response

This study reported the silver nanonets processed that have the features of nanoflakes and nanoparticles from porous silver nanoflakes as conducting fillers of flexible conductive films.

There are some comments to be made and covered as follows:

With respect to the abstract part, there is a lack of information and the novelty of the work needs more clarification. In addition, this part should be reviewed on the basis of the material, the objective, the main results obtained, as well as clarification of the work novelty.

Response: revised in the manuscript.

Keywords; the provided Keywords should be revised for example the author provids "ag flakes", moreover further keywords should be added related to the work tabout targeted, method or techniques, Ag nanonets......

Response: revised in the manuscript.

The Moreover, the provided claim at the end of the introduction needs more clarification and revision on the basis of the objective of the study, the material and approach, the relevance, the expected, and the key findings.

Response: revised in the manuscript.

4-      The experimental part:

i) can the author divide this part to "Materials and chemical” section

, and “Characterization and methods”., and 

Response: revised in the manuscript.

ii) The preparation method needs more clarification, in fact it is better to add scheme that clarify each step of preparation till the detection process. This means scheme 1 should be revised.

In fact during the results and discussion, the author described the figures only without clear discussion or interpretation. Therefore, it is highly recommended to give more information and details about the results obtained.

Moreover, the provided figure captions is not enough to describe the figure, the author must provide more information and details enough to clarify the figures.

The author should provide all figures with high resolution.

Clear results should be provided in the conclusion part.

Response: many revision have dong in the manuscript.

Reviewer 3 Report

The paper ”The Evolution Process from Silver Nanoflakes to Nanonets to Effectively Address the Trade-Offs of the Conductivity and Stretchability in Elastic Electrodes” deals with the problem of improving the conductivity of the flexible silver paste. Firstly, the Authors synthesized novel porous silver particles of several micromiters and then observed an intriguing transformation of the resulting particles into smoother but still porous flakes during solvothermal ripening in water. Finally, these flakes were used to prepare a flexible conductive films with a good mechanical properties and god conductivity robustness to stretching. The paper is worth publishing in Materials, but in its current form is unacceptable and requires a thorough revision.

1) I suggest changing the title to:

“From silver nanoflakes to silver nanonets: an effective solution to the trade-off between conductivity and stretchability in flexible electrodes”

or even to shorter

“From silver nanoflakes to silver nanonets: an effective trade-off between conductivity and stretchability of flexible electrodes”

2) “Preparation of elastic conductors” section in Experimental part should contain details so that scientists from other laboratories can repeat the chemical preparation. There is no specific data on chemicals. For example, wordings like “The elastic conductor pastes are comprised of Ag flakes, polyester rubber and some other additives” are unacceptable.

Please provide some details about the screen printing technique used, preparation and dried conditions (air or N2).

3) Figure 1b shows SEM image of as-obtained Ag flakes. However, the magnification is too small to observe the pores and thicknesses of the particles. Provide high resolution images in nm scale showing the thickness and size of the pores.

4) What amount of Ag powder was added to 10 mL of a solvent in solvothermal processing? Similar  to remark 3), provide a high resolution image for Figure 2a and 2d showing the thicknesses of the flakes and the size of the pores.

5) A number of other points need to be addressed as well.

p.5 - the term "stretching rate" concerns the change of deformation with respect to time. Here the Authors discuss stretching ratio or elongation.

p.6 - “anti-bending conductivity” there should be bending conductivity rather

Figure 5a. – “Sheet resistivity (mΩ/â–ˇ)”. Resistivity is measured in Ω m.

Figure 5b. – “Stretched length” - use “Elongation”

Figure 5b. – “Increased resistivity (%)”. It is inconvenient to express large changes in physical quantities in percent. It is better to apply relative changes in resistivity.

6) The manuscript should be thoroughly modified in terms of English to be accepted for publication. I did not undertake linguistic verification of the text.

Author Response

1) I suggest changing the title to:

“From silver nanoflakes to silver nanonets: an effective solution to the trade-off between conductivity and stretchability in flexible electrodes”

or even to shorter

“From silver nanoflakes to silver nanonets: an effective trade-off between conductivity and stretchability of flexible electrodes”

Response 1): The title have changed.

2) “Preparation of elastic conductors” section in Experimental part should contain details so that scientists from other laboratories can repeat the chemical preparation. There is no specific data on chemicals. For example, wordings like “The elastic conductor pastes are comprised of Ag flakes, polyester rubber and some other additives” are unacceptable.

Please provide some details about the screen printing technique used, preparation and dried conditions (air or N2).

Response 2): The title have changed.

3) Figure 1b shows SEM image of as-obtained Ag flakes. However, the magnification is too small to observe the pores and thicknesses of the particles. Provide high resolution images in nm scale showing the thickness and size of the pores.

Response 3): SEM image of thicknesses is provided in Figure 1b.

4) What amount of Ag powder was added to 10 mL of a solvent in solvothermal processing? Similar  to remark 3), provide a high resolution image for Figure 2a and 2d showing the thicknesses of the flakes and the size of the pores.

Response 4): Experimental part have been much revised in the manuscript.

5) A number of other points need to be addressed as well.

p.5 - the term "stretching rate" concerns the change of deformation with respect to time. Here the Authors discuss stretching ratio or elongation.

p.6 - “anti-bending conductivity” there should be bending conductivity rather

Figure 5a. – “Sheet resistivity (mΩ/â–ˇ)”. Resistivity is measured in Ω m.

Figure 5b. – “Stretched length” - use “Elongation”

Figure 5b. – “Increased resistivity (%)”. It is inconvenient to express large changes in physical quantities in percent. It is better to apply relative changes in resistivity.

Response: “mΩ/â–ˇ” is a kind of commonly used unit for thick film, and Ω m is commonly used unit for 3D material. mΩ/â–ˇ is used here should be acceptable.

Other suggestions have revised in the manuscript.

6) The manuscript should be thoroughly modified in terms of English to be accepted for publication. I did not undertake linguistic verification of the text.

Response: many revision have dong in the manuscript.

Reviewer 4 Report

The authors describe how different synthesized Ag nanoparticles can perform as fillers in stretchable rubber composites. 

Although the topic could be interesting and the particles synthesis is sufficiently argued, the rest of the experimental part must be improved with detailed and clear descriptions.

Moreover the whole paper needs a rewriting, because it's poor English makes it often confused.

In particular:

In the introduction it is not clear how this research will bring some novelty.

The experimental part lacks of details, especially the fabrication of the rubber samples and the characterization setup for measuring resistance. Also images of the samples are missing.

It is not clear how the authors obtained a enhanced conductivity and stretchability. Moreover 8 stretching cycles are not enough to characterize a stretchable conductor ageing.

In Figure 6 there is a sketch of a supposed dynamics of particles during bending but no experimental evidence is provided. Cross section SEM images of the composite also are absent.

The results of the experiments are not clearly described.

Author Response

Although the topic could be interesting and the particles synthesis is sufficiently argued, the rest of the experimental part must be improved with detailed and clear descriptions.

Moreover the whole paper needs a rewriting, because it's poor English makes it often confused.

In particular:

In the introduction it is not clear how this research will bring some novelty.

The experimental part lacks of details, especially the fabrication of the rubber samples and the characterization setup for measuring resistance. Also images of the samples are missing.

It is not clear how the authors obtained a enhanced conductivity and stretchability. Moreover 8 stretching cycles are not enough to characterize a stretchable conductor ageing.

In Figure 6 there is a sketch of a supposed dynamics of particles during bending but no experimental evidence is provided. Cross section SEM images of the composite also are absent.

The results of the experiments are not clearly described.

Response: many revision have dong in the manuscript.

Round 2

Reviewer 1 Report

Thanks for revising the paper. It looks good for publication.

Author Response

Thank the reviewers for their hard work and recognition of this article

Reviewer 2 Report

This work has been studied and published by the authors befor.

Please see this site

What is the novelty, and new?

The provided figures are the same.

Author Response

The preprints 201905.0375.v1%20(2).pdf is the production of this paper first to submission to Nano materials, and Nano materials reject this article. In general, publishers do not consider that a preprint of an article prior to its submission to publishers for consideration amounts to prior publication, which would disqualify the work from consideration for re-publication in a journal. We believe preprints is good to communicate scientific knowledge, scientific method and scientific spirit.

Reviewer 4 Report

The paper now is readable and many corrections have been made.

I point out that some errors are still present (e.g Chart page 7 "Resistivity" on the y axis)

The remaining point is:

The characterization setup for measuring resistance during bending should be descibed.

Moreover the quality of the paper would improve if cross section SEM images of the composite before and after stretching are added so I strongly suggest to add them ( if it is in the possibility of the authors ).

Author Response

I point out that some errors are still present (e.g Chart page 7 "Resistivity" on the y axis)

Response: the error have been revised in the manuscript.

The remaining point is:

The characterization setup for measuring resistance during bending should be descibed.

Response: The characterization methods for measuring resistance have been much supplemented in experimental part.

Moreover the quality of the paper would improve if cross section SEM images of the composite before and after stretching are added so I strongly suggest to add them ( if it is in the possibility of the authors ).

Response: We are very sorry. we once tried to observe the cross section of the conductive film by SEM. However, we find it is difficult to obtain the smooth and clear cross structure by mechanically cutting due to the ductility of silver. We will try again in the future work.
